# Low Buffer Trapping Effects above 1200 V in Normally off GaN-on-Silicon Field Effect Transistors

**DOI:** 10.3390/mi13091519

**Published:** 2022-09-14

**Authors:** Idriss Abid, Youssef Hamdaoui, Jash Mehta, Joff Derluyn, Farid Medjdoub

**Affiliations:** 1IEMN (Institute of Electronics, Microelectronics and Nanotechnology), Avenue Poincaré, 59650 Lille, France; 2SOITEC-Belgium N.V., Kempische Steenweg 293, 3500 Hasselt, Belgium

**Keywords:** high-electron-mobility transistor (HEMT), GaN, normally off

## Abstract

We report on the fabrication and electrical characterization of AlGaN/GaN normally off transistors on silicon designed for high-voltage operation. The normally off configuration was achieved with a p-gallium nitride (p-GaN) cap layer below the gate, enabling a positive threshold voltage higher than +1 V. The buffer structure was based on AlN/GaN superlattices (SLs), delivering a vertical breakdown voltage close to 1.5 kV with a low leakage current all the way to 1200 V. With the grounded substrate, the hard breakdown voltage transistors at V_GS_ = 0 V is 1.45 kV, corresponding to an outstanding average vertical breakdown field higher than 2.4 MV/cm. High-voltage characterizations revealed a state-of-the-art combination of breakdown voltage at V_GS_ = 0 V together with low buffer electron trapping effects up to 1.4 kV, as assessed by means of substrate ramp measurements.

## 1. Introduction

Among the semiconductors, wide band gap (WBG) materials such as GaN and SiC show more suitable properties than Si to operate at higher power and higher voltage [1,2,3,4,5,6]. In addition to being available up to 8 inches on Si substrates, allowing a cost-effective fabrication, GaN material offers high electron mobility via the formation of a two-dimensional electron gas (2DEG) at the hetero-interface between the AlGaN barrier and the GaN channel layer. These devices are inherently normally on, delivering a negative threshold voltage. In order to achieve a normally off behavior, several methods have been reported in the literature [7,8,9,10,11,12,13,14,15]. All of them involve depleting the 2DEG below the gate electrode. One of the most promising approach is the use of a p-GaN layer that is etched outside the gate contact region [16,17,18]. In this case, the p-GaN layer raises the band diagram, causing the depletion of the 2DEG even without external bias. On the other hand, although these types of devices are commercially available, they are limited to 650 V voltage operation. Buffer engineering is required to generate highly insulating transition layers grown on silicon and further enhance the voltage operation. The use of graded AlGaN buffer layers is the historical approach [19,20,21]. Al_x_Ga_1-x_N layers (several micrometers thick with different Al contents) allow the alleviation of lattice mismatch and thermal expansion between the silicon substrate and the GaN layer. Moreover, the introduction of intentional doping can significantly increase the resistivity of the buffer. Iron or carbon doping is generally used to produce highly resistive buffers by compensating the residual n-type doping, such as oxygen impurities, generally inducing parasitic leakage currents. However, it has been shown that a high Fe or C doping concentration generates electron trapping effects [22,23,24]. To suppress the undesired trapping effects while maintaining an excellent carrier confinement into the 2DEG under a high electric field, AlGaN back barrier [25] or superlattice buffer concepts [26,27] can be combined with a moderate C doping concentration. The desire to use AlN/GaN superlattices along with a carbon-doped buffer has been previously demonstrated for high-voltage capabilities in normally on transistors [28,29]. This concept involves a series of thin layers such as Al(Ga)N and GaN to avoid the formation of internal stresses while benefiting from highly resistive buffer layers.

In this paper, we experimentally fabricated normally off p-GaN cap AlGaN/GaN HEMTs using a superlattice buffer with a total thickness of 6 µm, combining a high blocking voltage at V_GS_ = 0 V and low trapping effects above 1 kV.

## 2. Materials and Methods

Figure 1 shows a schematic cross-section of the p-GaN/AlGaN/GaN heterostructures grown by metal organic chemical vapor deposition (MOCVD) on a 1 mm-thick 6-inch Si substrate. Following the AlN nucleation layer, a 6 µm total buffer thickness based on 140 periods of 25 nm AlN/GaN superlattice and a carbon-doped GaN layer of 5 × 10^18^ cm^−3^, an unintentionally doped GaN channel, a 12 nm Al_0.18_Ga_0.82_N barrier layer, and an 80 nm p-type doped GaN layer. The Mg concentration was 2 × 10^19^ cm^−3^.

The 2DEG properties obtained through Van der Pauw pattern showed an electron sheet concentration of 8 × 10^12^ cm^−2^ with an electron mobility of 1800 cm^2^/V·s. A Ti/Al/Ni/Au metal stack was deposited and annealed at 750 °C to form the source and drain ohmic contacts directly on top of the barrier layer by fully etching the p-GaN cap layer. Contact resistances of about 1 Ω·mm were obtained, which can be reduced with further optimization. The isolation between contacts was realized by mesa etching with a depth of 400 nm. Outside the gate area, the entire remaining p-GaN area was etched and a Ni/Au gate metal was deposited. Lastly, a 150 nm PECVD SiN layer was deposited as final passivation.

## 3. Results

Figure 2a shows the vertical breakdown voltage close to 1500 V corresponding to an excellent breakdown field higher than 2.4 MV/cm. This reflects the high crystal quality of the growth related to the superlattice concept. It can be pointed out that typical values for a similar buffer thickness are below 2 MV/cm.

Lateral breakdown voltage measurements between isolated ohmics contacts on various distances with the substrate floating confirm the high-voltage capabilities. In order to avoid arcing in air between the probes, the sample is immersed in a liquid solution (Fluorinert). As shown in Figure 2b, an expected linear evolution of the breakdown voltage as a function of the contact distances followed by saturation is observed. A significant lateral breakdown voltage up to 2500 V is measured for a contact distance of 16 µm and above. It can be noticed that a blocking voltage of about 2000 V is reached at 1 µA/mm.

Moreover, a key parameter to assess GaN-based for power switch devices is the trapping effects. Thus, buffer trapping has been studied by means of substrate ramp measurements. [30,31]. This measurement is essentially sensitive to the traps into the buffer and surface independent. Indeed, a low bias is applied within a TLM (ohmic contacts), generating a current while the substrate is biased from 0 V down to a high negative potential. Any charge redistribution into the buffer during the sweeps (back and forth) will be detected as a current change (hysteresis). Indications on the time constant of the traps can also be determined to a certain extent by varying the sweep rate.

Figure 3 shows different measurements down to 500 V, 1000 V, and 1400 V at various sweep rates of 4 V/s in blue and 22 V/s in black. Extremely low hysteresis is evident in all cases, reflecting the outstanding low trapping effects observed up to 1400 V, regardless of the sweep speeds. This is attributed to the excellent material quality and the low charge storage within the structure, enabling both a superior breakdown field and reduced buffer trapping.

Electrical characterizations are carried out on 2 × 50 μm transistors with a gate-to-drain spacing, ranging from 8 to 40 μm. Transfer characteristics I_D_-V_GS_, as shown in Figure 4 at V_DS_ = 4 V, reveal a low leakage current around 20 nA/mm and an excellent pinch-off behavior, showing the absence of parasitic punch-through effects or gate leakage current. Moreover, from the transfer characteristics plotted in semi-log and linear, a threshold voltage as high as +1.4 V is extracted, resulting in fully normally off transistors. Due to non-optimized ohmic contacts and a rather low Al content into the barrier, a moderate on-state current density of 150 mA/mm an on-state resistance (R_ON_) of about 32 mΩ/cm^2^ is obtained, as seen from the output characteristics in Figure 4.

Figure 5a shows the three-terminal off-state breakdown voltage of transistors at V_GS_ = 0 V as a function of the gate–drain distances with a floating and grounded substrate. The breakdown voltages of transistors with a grounded substrate are slightly higher than 1400 V, in agreement with the high breakdown field of the buffer layers. Similarly, as can also be seen in Figure 5b, the breakdown voltage with a floating substrate reaches 2500 V for large gate–drain distances higher than 20 µm. The leakage current remains below 1 µA/mm above 1 kV. The specific on-state resistance versus breakdown voltage with grounded substrate has been benchmarked against normally off GaN transistors in the literature (Figure 6) [32,33,34,35,36]. These results highlight the benefits of superlattice-based buffer, enabling high-performance normally off devices with low R_ON_ and low buffer trapping at a blocking voltage above 1200 V.

## 4. Conclusions

In summary, this paper demonstrated the fabrication and characterization of a state-of-the-art normally off AlGaN/GaN heterostructure with a buffer based on superlattice structure capped with a p-GaN layer. This resulted in normally off transistors with a threshold voltage higher than +1 V. High vertical breakdown voltage close to 1500 V can be reached with low trapping effects. The three-terminal breakdown voltage measurements carried out on transistors show a breakdown field higher than 2.4 MV/cm with a grounded substrate and a breakdown voltage of 2500 V with floating substrate. These results unveil the excellent prospects of a superlattice-based buffer for 1200 V power applications, such as fast chargers for electric vehicles, motor drives, solar inverters, or three-phase PFC systems.

## Figures and Tables

**Figure 1 micromachines-13-01519-f001:**
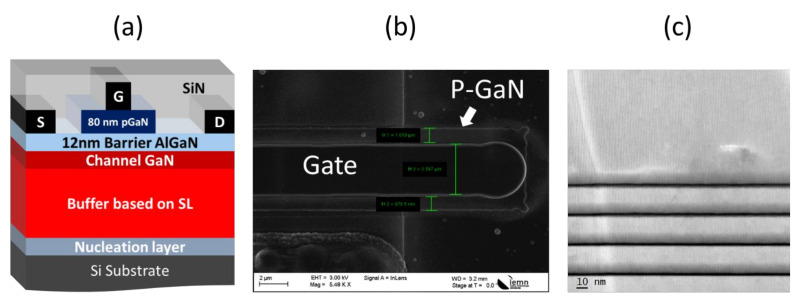
(**a**) Schematic cross section of the p-GaN/AlGaN/GaN HEMT on Si. (**b**) SEM top view of the gate including the partially etched p-GaN layer and (**c**) a zoomed TEM image of the superlattices.

**Figure 2 micromachines-13-01519-f002:**
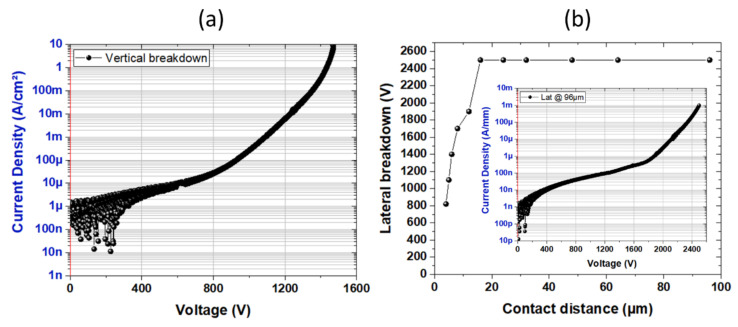
Vertical breakdown voltage (**a**) and lateral floating breakdown voltage as a function of the contact distances (**b**) and the p-GaN/AlGaN/GaN HEMT on Si at room temperature.

**Figure 3 micromachines-13-01519-f003:**
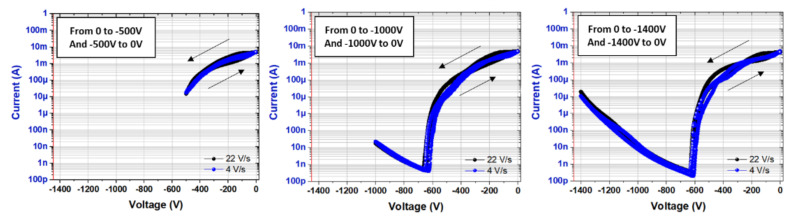
Substrate bias ramp measurements performed on the p-GaN/AlGaN/GaN HEMT on Si at room temperature.

**Figure 4 micromachines-13-01519-f004:**
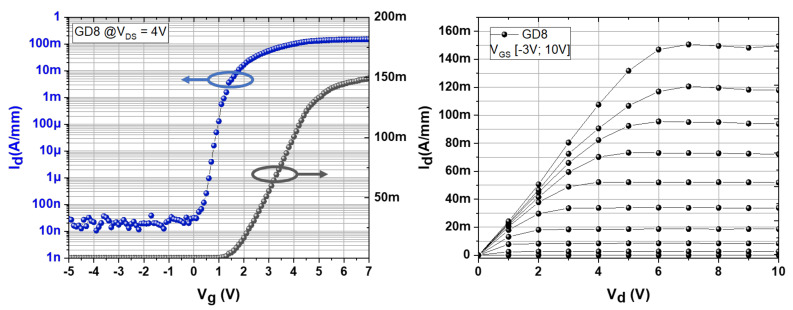
Transfer (**left**) and output (**right**) characteristics of a p-GaN/AlGaN/GaN HEMT with a gate–drain distance of 8 µm at V_DS_ = 4 V plotted in semi-log and linear scale.

**Figure 5 micromachines-13-01519-f005:**
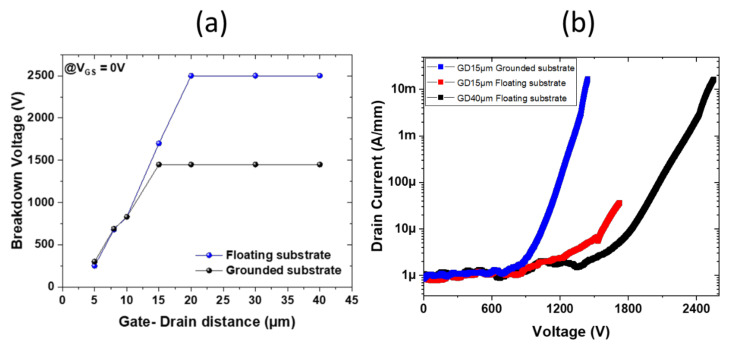
Three-terminal breakdown voltage at V_GS_ = 0 V of p-GaN/AlGaN/GaN HEMTs with floating and grounded substrate: (**a**) as a function of the gate-to-drain distance and (**b**) for various gate-to-drain distances.

**Figure 6 micromachines-13-01519-f006:**
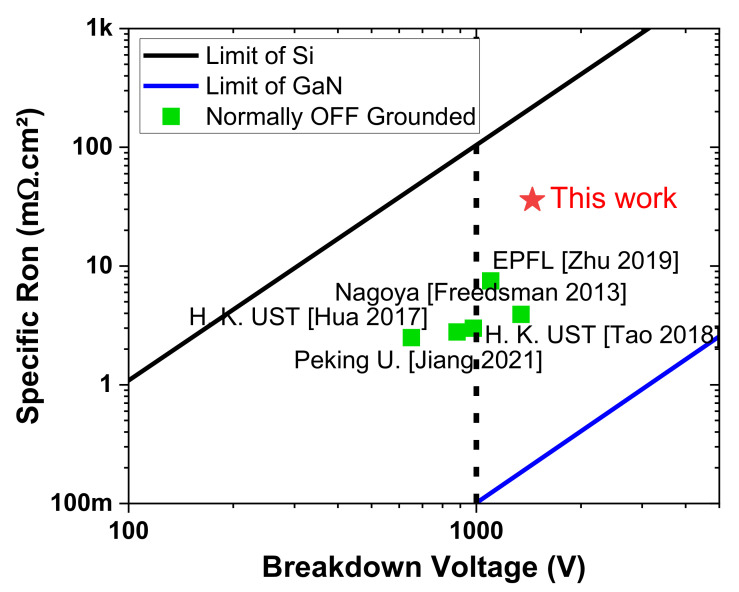
Benchmarking of breakdown voltage vs. specific on-resistance of normally off transistors with a grounded substrate. Solid lines in black and blue represent the theoretical limits of Si and GaN, respectively.

## Data Availability

Not applicable.

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
