# Peer review of "Low Buffer Trapping Effects above 1200 V in Normally off GaN-on-Silicon Field Effect Transistors"

_micromachines, 2022, doi:10.3390/mi13091519_

Round 1

Reviewer 1 Report

The paper presents the fabrication and characterisation of an AlGaN/GaN on silicon normally-off high electron mobility transistor.  The Paper is well written and easy to understand.

Authors can add to the manuscript the information on possible applications for analysed transistors. It is not known if the breakdown voltage of the final transistor could be 1200 V or 1500 V? 

Author Response

Reviewer 1

Authors can add to the manuscript the information on possible applications for analysed transistors. It is not known if the breakdown voltage of the final transistor could be 1200 V or 1500 V?

The authors would like to thank the reviewer for the time spent on our paper and his valuable comments. Please find below our point-by-point responses to the comments:

Examples of 1200V applications include fast chargers for electric vehicles, motor drive, solar inverters or three phase PFC systems. This has been added at the end of the conclusion.

As mentioned in Fig. 5, the hard breakdown voltage of the final transistors at VGS = 0V is 1450V for a gate-drain distance of 15 µm and above.

Reviewer 2 Report

The paper presents a normally off AlGaN/GaN transistor achieved through the use of a thick superlattice layer to both act as the high resistance barrier and trap prevention at high voltage applications.  The paper is well written and the experimental methodology is sound. The presentation of the results are excellent. The language style is clear. The paper can be considered for publication in micromachines after the following minor changes can be added in the manuscript:

1. Figure 1 should include a zoomed in diagram of at least one period of the superlattice layer structure of alternating Al/GaN C-doped GaN layer to clarify to the general reader what the supperlattice looks like. 

2. Mention in the manuscript the reason for using 140 periods of the superlattice layer and is this the optimized value to achieve the desired resistivity value of the layer. 

3. Mention in the manuscript what the approximate doping concentration of C in the C-doped GaN layer.

Author Response

The authors would like to thank the reviewer for the time spent on our paper and his valuable comments. Please find below our point-by-point responses to the comments:

The paper presents a normally off AlGaN/GaN transistor achieved through the use of a thick superlattice layer to both act as the high resistance barrier and trap prevention at high voltage applications.  The paper is well written and the experimental methodology is sound. The presentation of the results are excellent. The language style is clear. The paper can be considered for publication in micromachines after the following minor changes can be added in the manuscript:

  1. Figure 1 should include a zoomed in diagram of at least one period of the superlattice layer structure of alternating Al/GaN C-doped GaN layer to clarify to the general reader what the supperlattice looks like. 

As advised by the reviewer, we included a zoomed TEM image of the superlattice in Fig. 1.

  1. Mention in the manuscript the reason for using 140 periods of the superlattice layer and is this the optimized value to achieve the desired resistivity value of the layer. 

We started in a previous with a lower number of pairs that has been increased to 140 in order to reach a thicker overall thickness while achieving reduced strain management and flat wafer. However, we do not yet know if this value is the limit (in terms of strain). Additional experiments would be needed to check the possibility to further increase the number of pairs.

  1. Mention in the manuscript what the approximate doping concentration of C in the C-doped GaN layer.

The approximate value of C-doping is 5×1018 cm-3, which has been added in the manuscript.

Reviewer 3 Report

The authors demonstrated normally-off GaN on Si transistor with improved breakdown voltage. The manuscript is worthy of publishing in Micromachines. The following points should be addressed.

1. The authors have attributed to several merits of their devices to the superlattice used in the structure. However, there's no experimental evidence on the high quality, it will be helpful to readers if the XRD, TEM data can be included.

2. Do the authors have tried any control experiment to show that the improvement in the device performance is from the superlattice buffer?

3. What's the estimated over-etching into AlGaN barrier during the p-GaN etch?

4. There are inconsistent font style among the figures.

Author Response

Reviewer 2

The authors would like to thank the reviewer for the time spent on our paper and his valuable comments. Please find below our point-by-point responses to the comments:

  1. The authors have attributed to several merits of their devices to the superlattice used in the structure. However, there's no experimental evidence on the high quality, it will be helpful to readers if the XRD, TEM data can be included.

As referenced in the paper, we previously reported on an electrical comparison of the superlattice-based buffer with a more standard step-graded buffer "Low On-Resistance and Low Trapping Effects in 1200 V Superlattice GaN-on-Silicon Heterostructures", Physica Status Solidi (a) 2019, 217, 7, 1900687. In this work, we developed fully normally-off transistors based on this technology and used a slightly thicker total buffer thickness as compared to the previous run in order to further enhance the breakdown voltage. Furthermore, substrate bias ramp measurements using various sweep rates at room temperature uniformly delivers very low trapping effects all the way to 1400 V regardless of the sweep rates, reflecting the high crystal and growth quality.

In addition, we can see on the attached file some TEM images of the buffer and more particularly of the super-lattices that we did not include because of space limitations.

  1. Do the authors have tried any control experiment to show that the improvement in the device performance is from the superlattice buffer?

Please check the following paper from our group showing the comparison with a control structure (standard step-graded buffer): "Low On-Resistance and Low Trapping Effects in 1200 V Superlattice GaN-on-Silicon Heterostructures", Physica Status Solidi (a) 2019, 217, 7, 1900687.

  1. What's the estimated over-etching into AlGaN barrier during the p-GaN etch?

Following extensive tests and non-selective etching optimisations, it is estimated that the barrier layer etching is between 1 to 2 nm.

  1. There are inconsistent font style among the figures.

This has been corrected in the paper.
